# Burnout and Resilience in Foreign Service Spouses during the Pandemic, and the Role of Organizational Support

**Svala Gudmundsdottir** [1],*[ID]**, Karen Larsen** [2][ID]**, Melissa Woods Nelson** [2][ID]**, Jarka Devine Mildorf** [2]
**and Dorota Molek-Winiarska** [3]

[1]   Department of Business Administration, School of Social Sciences, Gimli Sæmundargata,
      University of Iceland, 102 Reykjavík, Iceland
[2]   EUFASA Research Department, EUFASA AISBL, Boulevard Bischoffsheim 39-4, 1000 Brussels, Belgium
[3]   Human Resources Management Department, Faculty of Management, Wroclaw University of Economics and
      Business, Komandorska 118/120, 53-345 Wroclaw, Poland
*    Correspondence: svala@hi.is

**Abstract:** Little is known about foreign service spouses' resilience and experience with stress, or which organizational supports are most effective for them in a crisis. The aims of this study were therefore to (1) measure resilience and personal burnout among foreign service spouses during the COVID-19 pandemic; (2) test whether resilience was associated with personal burnout; and (3) investigate which organizational supports from Ministries of Foreign Affairs (MFAs), if any, were associated with resilience or personal burnout. A total of 421 foreign service spouses (316 women and 105 men, 89% from European MFAs) completed an online survey; data were analyzed using multiple regression analyses. Nearly a third of respondents (31%) had moderate personal burnout and 4.5% had high or severe personal burnout. Higher resilience was significantly correlated with lower personal burnout scores. While knowledge that they would be evacuated if necessary was significantly correlated with greater resilience, only 27.3% of respondents knew of this support. A designated contact person at the MFA, and knowledge of a policy abroad to reduce Covid-related risks were associated with lower personal burnout scores. This study provides a baseline for resilience and personal burnout in this population, and suggests which organizational supports may be most effective during a crisis.

**Keywords:** diplomatic spouse; foreign service spouse; expatriate; foreign service; resilience; burnout; organizational support; expatriate management

## 1. Introduction

Expatriation has been widely demonstrated to be stressful [1–5], and McNulty [6] pointed out that at least half of the 40 most stressful life events can be directly or indirectly related to relocation of families. Several previous studies have demonstrated negative health consequences among expatriate workers including burnout [7–11] and consequences for psychological well-being and family relationships [12,13].

Foreign service officers (we use the term "foreign service officer" in this paper as an inclusive term for all transferable Ministry of Foreign Affairs employees, as in some foreign service systems, diplomatic and consular careers are separate, and not all foreign service officers are diplomats) and their families are a distinct subgroup of expatriates who are typically transferred every three to five years between locations that are important to achieve the diplomatic, consular, and/or development objectives of the sending state [1,14]. While there are differences across foreign services, the pattern of regular international transfers is often maintained until retirement. The posting locations expose foreign service officers and their families not only to culture shock, loss of social support structures, and language challenges, but also sometimes to political strife, increased security risks, high poverty rates and social inequality, poor health systems, and other stressors. Foreign

service spouses (we use the term "foreign service spouses" to refer to all partners and spouses who accompany foreign service officers on overseas postings, whether they are married or not) are often unable to maintain their own employment due to the regular transfers and are sometimes also expected to assist the foreign service officer in hosting and attending official events [15–18]. Despite the attention given in the literature to private-sector expatriate workers, the health-related challenges faced by foreign service officers and their partners/spouses have rarely been addressed in research studies [3,14].

The COVID-19 pandemic produced tremendous uncertainty and stress, affecting the mental health of many around the world [19]. For expatriates, further stressors were at play: not only did they face the risk of contracting a serious new illness, but they were doing so while far from their usual social supports, perhaps while living in a country with a poor health system and/or where they did not speak the language, while also worrying about loved ones at home they could not visit due to travel restrictions. These factors may have contributed to high levels of stress among expatriates during the pandemic, as found, for example, by Haist and Kurth [20].

Previous studies have highlighted the importance of organizational support to foster expatriate adjustment, and have reported the outsize role of spousal factors in expatriate failure [6,17,21]. However, only a handful of studies have looked at accompanying spouses empirically, and even fewer have examined foreign service spouses. No published studies to date appear to have addressed either foreign service spouses' resilience and experience with stress, or organizational supports to support their mental well-being. Given that Ministries of Foreign Affairs (MFAs) depend upon their foreign service officers to remain internationally mobile, and that their partners and spouses play a critical role in the success of their placements abroad, good support of partners and spouses is essential for MFAs to achieve their goals. Understanding resilience and stress, as well as which types of organizational support appear to be helpful for supporting partners' and spouses' mental health, is therefore important for MFAs, especially during a crisis.

## 2. Theoretical Background and Research Aims

The understanding of burnout has evolved since its identification, and differences in the definition and measurement exist; however, it is most widely understood as a reaction to chronic stress which results in exhaustion, feelings of detachment or depersonalization, and a sense of inefficacy [22–24]. Several models have been proposed for how burnout develops, for example, organizational theory, which considers burnout a result of work and organizational stressors, combined with inadequate coping strategies [25], and the job demands–resources model, which suggests that burnout may result when job demands outweigh the physical, psychological, or organizational resources one receives from the job [26]. Research has indicated that many factors can contribute to the development of burnout on the macro level (such as changing values or global economic shifts [27]; on the organizational level (poor person–organization fit, stressful environment, dysfunctional relations, etc.); and on the individual level (for example, low self-esteem, low resilience, and poor communication skills [27–29].

Although burnout was first described in professionals involved in emotionally taxing work, such as nurses, physicians, and teachers, understanding of the phenomenon has since broadened. It was later found to occur in other professions when there is a lack of fit between the person and the organization, in marriage or partnership [30], and among both patients suffering from longitudinal illness, as well as their spouses [31]. Fatigue and exhaustion resulting from long-term involvement in emotionally demanding situations are at the core of burnout, which may occur in both traditional work and non-work conditions [32,33]. Burnout is, therefore, not specific to an employment environment; "personal burnout" refers to the degree to which physical and psychological fatigue and exhaustion are experienced, regardless of occupational status [34].

Therefore, not only can foreign service spouses who work in the traditional sense experience burnout; the roles which are placed upon them as foreign service spouses may

also contribute to stress. Regular moves, loss of employment opportunities and social supports, culture shock, the inability to communicate in the local language, and/or other demands may also contribute to emotional and physical exhaustion in some spouses. The COVID-19 pandemic is likely to have added further strains on this population. Burnout among spouses would not only affect them personally; the spouse's mental well-being can also affect the employee's effectiveness, as stressors experienced in the family can have negative spillover effects on the expatriate performance at work [35]. Spousal factors are also well-established as important causes of expatriate failure [6,21].

Resilience has been described as the capacity to adapt and "bounce back" in the face of crisis or adversity [36–38] or the ability to "harness resources to sustain well-being" [39]. Chen et al. conceptualize resilience as a combination of self-esteem and self-efficacy [40]. Some have considered resilience to be a skill or individual ability to adapt to adverse situations [41]. In this context, resilience may be treated as a permanent resource in coping with stressful environments. Others suggest that resilience is a unique, complex process of adapting to an adverse environment. Three different models have been proposed for the mechanism. The first, a compensatory model, assumes that protective factors compensate for the hazardous environment [42]. For example, stressful situations at work are made easier by a supportive coworker or partner. In the second, an immunity or protective model, protective factors interact with risk factors to reduce their negative influence on the individual [43]. For example, good informational support during a reorganization process may reduce the stress caused by the changes. The third, a challenge model, implies that a moderate level of stress may test and "toughen up" a person to face more difficult situations [44]. Resilience has been found to be related to the individual's sense of control over the situation [39,45], and to change over time [46,47]. Studies have found that resilience can be increased in a number of ways, for example, through social connectedness and selfcare [48,49]; exercise and nutrition [50]; work-based programs aiming to increase self-efficacy, optimism, resources and coping skills [51]; social support and bonding [52]; helpful information [48]; and clear communication in and between institutions [53,54].

Resilience has been found to counteract or mitigate burnout by reducing levels of anxiety, depression, and other negative psychological symptoms [55–57]. While workplace norms and perceived and actual work expectations may be influenced by culture, low resilience has been found to be significantly correlated with burnout in a variety of stressful settings and cultural contexts, including among Japanese nurses [58]; Korean call center workers, school counsellors, and mental health workers [59]; child protection workers in Northern Ireland [60]; US physicians [61]; and others. On the other hand, high resilience may help individuals to not only make it through traumatic events, but even grow and learn from the experience in what has been called post-traumatic growth [62,63]. Abram and Jacobovitz argue that resilience protects individuals from the psychological consequences of burnout, but may not always prevent it [64]. Although an inverse relationship between resilience and burnout has been found in many studies it is not a one-way relation; both variables may influence the other. Given its apparent protective role, resilience is an important phenomenon to understand in stress reactions, including burnout.

Critically, all three of the above models of resilience demonstrate the crucial role of support in the environment to increase resilience and protect against burnout. Various types of support, such as social networks, emotional support, information and communication, as well as material support, have been found to be correlated with burnout and resilience [39,65,66]. Conversely, resilience without support may leave a person feeling alienated or marginalized from resources that should be provided [67].

The need for better organizational support of expatriate spouses has long been recognized to improve staff mobility and to prevent premature termination of foreign assignments [1,16,68,69], but little is known about foreign service spouses. In an environment of repeated international relocation with resulting employment disruption, lost social connections, and linguistic and cross-cultural challenges, we assume that stress levels may be high among foreign service spouses, particularly during a crisis. Resilience may therefore be an

important resource for the foreign service spouse, and more knowledge on how Ministries of Foreign Affairs can best support resilience and prevent burnout may be helpful. Based on the above, our research therefore had the following aims:

A.   to measure levels of resilience and personal burnout among foreign service spouses during the COVID-19 pandemic;
B.   to test whether increased resilience was associated with a reduced risk of personal burnout in this population; and
C.   to investigate which forms of organizational support from Ministries of Foreign Affairs, if any, were associated with either greater resilience or a reduced risk of personal burnout.

## 3. Materials and Methods

### 3.1. Survey Instrument

As not all foreign service spouses are able to work in remunerative positions, we used only the personal burnout subscale of the Copenhagen Burnout Inventory (CBI). This 6-item subscale includes questions such as, "How often are you physically exhausted?", "How often do you think: 'I can't take it anymore'?", and "How often do you feel worn out?" Responses to each question are on a 5-point Likert scale ranging from "always/to a very high degree" to "never/to a low degree" [34]. The personal burnout score is obtained by averaging the scores from each of the corresponding 6 questions for each respondent, resulting in a score between 0 (no personal burnout) to 100 (severe personal burnout).

Resilience was assessed with the 10-item Connor–Davidson Resilience Scale (CD-RISC). Items in the scale include statements such as "I am able to adapt to change," "I think of myself as a strong person," and "Coping with stress can strengthen me." Each item is scored on a 5-point Likert scale such that a respondent's total score can range from 0–40, with higher scores indicating greater resilience [36,55]. This abbreviated scale has been validated in a variety of populations and languages [70–72].

Organizational support was measured by giving respondents a list of possible supports, and asking them to indicate all of the supports their MFA provided, if any. The list included supports such as access to a contact person or Family Office at headquarters; workshops/tips on stress management/resilience; access to a psychologist/counseling; clear policy/measures taken by the representation to reduce the risks of contracting COVID-19 at the embassy/mission and official housing; and the possibility of evacuation by the MFA if necessary. Respondents were also able to indicate other supports which were not on the list. Additional questions measured demographics, changes in workload, and how much respondents were personally affected by the pandemic.

### 3.2. Target Population

To reach a wide and broadly representative group of foreign service partners and spouses, the member associations of the European Union Foreign Affairs Spouses, Partners, and Family Association (EUFASA) were asked to distribute the anonymous online questionnaire among their members. During the study period, EUFASA had 19 member states: Austria, Belgium, Czech Republic, Estonia, the EU, Finland, France, Germany, Iceland, Ireland, Italy, Latvia, Lithuania, Luxemburg, Poland, Portugal, Spain, Switzerland and the UK. The survey was also disseminated via some European Ministries of Foreign Affairs and in an online group for foreign service spouses.

### 3.3. Data Analysis

Data were collected anonymously via Google forms in December 2020–January 2021, and were analyzed using Excel and R. Inspection of the correlation matrix revealed only a few coefficient values over 0.3 and none higher than 0.41. The Kaiser–Meyer–Olkin value was 0.7, which more than meets the recommended value for factorability of the correlation matrix [73–75]. Burnout and resilience were measured using multiple questions on scales. To assess internal consistency of the measures, Cronbach's alpha coefficients were calculated

showing high internal consistency for both burnout ($\alpha = 0.88$) and resilience ($\alpha = 0.87$). Hierarchical multiple regression analyses were conducted for the dependent variables on the independent variables including demographics, and support. Unstandardized regression coefficients were chosen to ease interpretation of results. Variance inflation factor analyses were run to assess levels of multicollinearity, with no inflation factor reaching above 2.0. Regression analyses controlled for age, female gender, number of international moves as an adult, and the number of children under 16 in the household.

## 4. Results

### 4.1. Demographics of Respondents

A total of 421 self-selected respondents completed the survey. Most (89.8%) respondents were spouses of foreign service officers from Europe; 3.0% were from Asia/Oceania; 2.5% were from Africa and the Middle East; 2.3% from the USA; and 2.5% other North and South American countries. Three-quarters (75%) were female and 35.5% had been born in a different country than their foreign service officer spouse. Ages were well distributed among respondents, with a median age in the mid-to-late 40s. More than a quarter (27.8%) reported being the spouse of a Head of Mission (generally an Ambassador or Consul General). About a fifth (21.4%) had one child age 15 or younger at home; a quarter (24.7%) had two children, and 8.3% had three or more children at home. Nearly one half (45.6%) of all respondents reported having no children under age 16 living with them. As expected, respondents were highly mobile; they reported having moved internationally an average of 5.4 times in their adult lifetimes. One quarter (25.7%) had been at the current location less than a year, about two-fifths (39.5%) more than one but less than three years; and 22.2% for three to four years. Few respondents (12.8%) reported having been at the current location for more than four years.

### 4.2. Personal Burnout

Our respondents appeared to experience high rates of personal burnout during the COVID-19 pandemic. The average personal burnout score among respondents was 41.53 (SD = 19.27). Nearly a third (31%) of spouses appeared to have moderate personal burnout (score 50–74); 3.3% had scores indicating high (score 75–99); and 1.2% had scores indicating severe (score 100) personal burnout as defined in previous research [76].

Personal burnout was found to decrease with age, and to increase with the number of school-age children at home (see model 2 Table 1). For each year of age, burnout fell roughly 0.57 points, decreasing about 11 points between a 30-year-old spouse and a 50-year-old spouse. Each child under 16 in the household increased a respondent's burnout score by 2.4 points. Female spouses had burnout scores which were 4 to 5 points higher than male spouses, and each international move increased burnout by nearly half a point ($F_{(4, 416)} = 11.555$, $p < 0.000$, $R^2 = 0.1$).

**Table 1.** Results of regression analyses of burnout and resilience.

| Dependent Variable | Resilience | Burnout | Burnout | Burnout | Burnout |
|---|---|---|---|---|---|
| Covariates | Model 1 β (SE) | Model 2 β (SE) | Model 3 β (SE) | Model 4 β (SE) | Model 5 β (SE) |
| Constant | 27.558 *** (0.517) | 59.306 *** (5.823) | 61.433 *** (5.909) | 75.955 *** (6.535) | 20.57 *** (14.789) |
| Female | | 4.637 * (2.109) | 5.145 * (2.09) | 4.504 * (2.049) | 1.1562 * (2.303) |
| Age | | −0.568 *** (0.122) | −0.565 *** (0.122) | −0.553*** (0.119) | −0.141*** (−4.85) |

**Table 1.** *Cont.*

| Dependent Variable | Resilience | Burnout | Burnout | Burnout | Burnout |
|---|---|---|---|---|---|
| Number of children under 16 in household | | 2.441 ** (0.881) | 2.224 * (0.875) | 2.207 * (0.857) | 0.575 ** (2.749) |
| Number of international moves as an adult | | 0.492 * (0.231) | 0.443 (0.229) | 0.546 * (0.224) | 0.126 * (2.29) |
| I can ask questions to our MFA/Family Office | 0.889 (0.998) | | −5.977 * (2.55) | | |
| Evacuation if needed | 1.668 * (0.809) | | 0.158 (2.089) | | |
| Helpful information for spouses/partners on a website or newsletter | −0.359 (1.07) | | −1.580 (2.729) | | |
| Clear policy/measures taken by the representation to reduce the risks of contracting COVID-19 at the mission/in residences | −0.101 (0.723) | | −4.190 * (1.849) | | |
| Social events / social support, also online | 0.8 (1.222) | | −3.982 (3.122) | | |
| Access to psychologist/counselling | −1.015 (1.02) | | 2.961 (2.617) | | |
| Workshop/course/tips on stress, coping skills, or resilience | −2.18 (1.285) | | 7.949 * (3.288) | | |
| Resilience | | | | −0.624 *** (0.123) | |
| Moving home | | | | | −2.136 ** (−2.791) |
| $R^2$ | 0.021 | 0.100 | 0.138 | 0.153 | 0.1167 |
| N | 421 | 421 | 421 | 421 | 421 |
| F | (7,413) = 1.238 | (4,416) = 11.555 | (11,409) = 5.953 | (5,415) = 14.973 | (5,415) = 10.963 |
| Significance | 0.281 | <0.000 | <0.000 | <0.000 | <0.000 |

\* Significant at $p < 0.05$, \*\* Significant at $p < 0.01$, \*\*\* Significant at $p < 0.001$.

Moving back to headquarters during the pandemic was associated with significantly lower (β = −2.14, $p < 0.0055$) personal burnout scores (F(5,415) = 10.963, $p < 0.000$, $R^2$ = 0.1167. See model 5, Table 1).

### 4.3. Resilience

The average resilience score among respondents was 27.7 as measured with the 10-item Connor–Davidson Resilience Scale. More experienced spouses (those who had moved more often, as well as spouses of more senior foreign service officers) were not found to be more resilient than those with less experience.

### 4.4. Correlation of Resilience with Personal Burnout

Multiple regression analysis was performed to test the effect of resilience on personal burnout, controlling for gender, age, the number of children under 16 in the household, and the number of international moves as an adult (F(5,415) = 14.97, $p < 0.000$, $R^2$ = 0.153). We observed a highly significant inverse correlation between burnout and resilience (β = −0.624, $p < 0.000$), meaning that for each 1-point increase in resilience, personal burnout scores fell 0.624 points (see model 4, Table 1).

*4.5. Organizational Support*

Nearly three quarters of respondents reported having some form of organizational support from their sending MFA. About two fifths of respondents (40.6%) said that their MFA had a clear policy/measures to reduce risk due to COVID-19, and 27.3% knew they could be evacuated if necessary. Only 21.4% said that they have a contact person or Family Office at headquarters and less than one fifth (19%) said they had access to helpful information for partners/spouses. Only 17.1% of respondents stated that they had access to psychological support services, and more than a quarter (26.1%) reported not having any support from their MFA at all.

To see which organizational supports, if any, were associated with greater resilience, we ran a multiple regression model (see model 1 in Table 1). The only organizational support that was significantly associated with resilience was knowing that the MFA would evacuate the respondent if necessary; those who stated that their MFA would evacuate them, if necessary, had resilience scores which were 1.7 points higher (($\beta = 1.668$, $p < 0.040$) than those who were not aware of such support ($F_{(7,415)} = 1.238$, $p < 0.281$, $R^2 = 0.021$).

Additional regression analyses were conducted to assess which organizational supports, if any, appear to have an effect on burnout. Having a contact person at the MFA ($\beta = -5.977$, $p = 0.02$) and having clear policy /measures to reduce risk ($\beta = -4.190$, $p = 0.024$) were both significantly correlated with personal burnout. Respondents who stated that they could ask someone at their MFA questions had burnout scores which were nearly 6 points lower than those who did not. Respondents who said that their MFAs provide clear information and policy to reduce risk had burnout scores approximately 4 points lower than people who did not. Interestingly, the provision of workshops, courses or information on stress management and resilience was positively correlated with burnout ($\beta = 7.949$, $p = 0.016$) ($F_{(11,409)} = 5.953$, $p < 0.001$, $R^2 = 0.14$; see model 3 in Table 1). Respondents who were aware that workshops were available to them had burnout scores which were nearly 8 points higher than people who were not. Other measures had no significant effect.

## 5. Discussion

This paper reports the first results on burnout and resilience among foreign service spouses that we are aware of. Our results indicate that this population experienced high stress during the pandemic, but that some forms of organizational support appear to have a protective effect on partners' and spouses' mental health during the pandemic.

The high average personal burnout score found in this study indicates significant distress in this population and should raise concern. Personal burnout scores among respondents were slightly higher than among many high-stress professions studied before the pandemic, including prison wardens, social workers, and hospital doctors and nurses [34]. In fact, of 15 high-stress professions studied by Kristensen et al., only two (midwives and home care assistants) were found to have higher personal burnout scores than was observed among our respondents [34]. We assume that personal burnout scores in this population would have been lower in the absence of the pandemic, but as no baseline data exist among foreign service spouses, and as Kristensen et al. did not assess burnout during a crisis, comparison is difficult. However, as 80% of our respondents indicated that their workload (whether employment- and/or household-related) had increased due to the pandemic (not shown), this suggests that at least some of the burnout measured was due to or exacerbated by the pandemic.

We expected more experienced foreign service spouses to be more resilient than less experienced spouses, as suggested by the challenge model of resilience. However, respondents with more experience (those who had moved more often, as well as spouses of more senior foreign service officers) were not found to be more resilient than those with less experience. This suggests that the challenge model may not be an appropriate model of resilience in this population, or perhaps that, on average, compensatory or

protective mechanisms of support have not been strong enough to overcome the challenges of international transfers and to build resilience through experience.

The strong inverse correlation observed between resilience and personal burnout in this study supports the theory that resilience plays an important role in preventing burnout among foreign service spouses, and agrees with findings in other populations. Other studies during the COVID-19 crisis have also reported a significant association between burnout and resilience in diverse populations, for example, among Dutch intensive care doctors [77], Indian and Chinese nurses [78,79], Polish medical students [80], and parents in many countries [81].

Our research suggests that some forms of organizational support from MFAs may bolster resilience and reduce the risk of personal burnout among foreign service partners and spouses. Interestingly, knowledge of the possibility of evacuation was the only organizational support which appeared to support resilience. This may be a direct or indirect result of, for example, communication by MFAs, spouses' level of agency in seeking information, and/or the level of trust respondents have in their MFA, among other factors. Unfortunately, only a minority (27.3%) knew of this support.

Having a designated contact person or Family Office at the MFA, and knowledge of a policy or measures to reduce the risk of developing COVID-19 were significantly associated with reduced personal burnout scores. However, respondents who reported that their MFA had offered a workshop or tips on stress management or resilience had significantly *higher* burnout scores than those who did not. We assume that this may be because stress management strategies are often preventive in nature, implying that they are best learned and practiced *before* a crisis. Rather than concluding that such supports are counterproductive, therefore, we assume that this is an indicator of the severity of the situation, and that these supports should be offered before a crisis strikes, and not when stress levels are already extremely high.

Previous research has highlighted the importance of good organizational communication for employees' mental health; Leiter and Maslach [82] described strategies to reduce the risk of burnout in an organization by providing clear and accessible information, immediate feedback, the possibility to receive explanations when necessary, and mutual trust. Similarly, Kim and Lee [83] and Atouba and Lammers [84] found that clear and supportive communication inhibit the development of burnout. The need for good communication during a crisis such as COVID-19 has also been well documented [85–89].

Several findings in our study, however, indicate that communication with partners and spouses needs improvement. Respondents who said that they had a contact person for information at their MFA were aware of more support measures offered by their MFA than those who said they had no contact person, suggesting that a designated contact person/Family Office may play a key role in providing information to partners. However, only one fifth (19%) of respondents said that they had access to helpful information for partners/spouses, and only one quarter (25%) of spouses felt they were receiving enough information from their MFA (not shown). We also assume that, if truly necessary, most, if not all, MFAs would evacuate their expatriate staff and families, and that the low percentage of respondents who were aware of this support therefore indicates a need for better communication with spouses, rather than an actual lack of evacuation support. MFAs may hesitate to communicate the supports they offer in order not to raise expectations, but our data suggest that simply knowing about a support (in this case, evacuation) may improve resilience.

Our research has several management implications for MFAs. First, MFAs should be aware that foreign service partners and spouses can develop personal burnout during a crisis; adequate supports and information are, therefore, essential not only to ensure the well-being of these family members, but also to allow foreign service officers to focus on their work. Secondly, based on our results, we recommend that important information and measures of support be actively communicated directly to partners and spouses, for example, through a Family Office or other direct contact person within the MFA, particularly

during a crisis when information is needed quickly. Direct communication with spouses can be a challenge for MFAs without a designated contact person; creative solutions, such as providing spouses with access to a chatroom or an app that gives spouses emergency information from the MFA, might help facilitate better communication. Finally, our results suggest that it is important to provide adequate psychological support to foreign service partners and spouses.

*Limitations and Future Research*

Foreign service spouses are a highly heterogenous group with a wide range of needs. Although the basic demographics of our respondents were similar to previous studies in this population, we cannot assume that our sample is fully representative of foreign service spouses in European systems or in general. Indeed, not only do the specific conditions of international transfers and organizational support vary between the foreign services of different countries, even within Europe; variations also occur within systems, for example, by posting location and the foreign service officer's level of seniority. However, our respondents all shared the (otherwise unusual) experience of regular international transfers as a foreign service spouse, and our results suggest that resilience and burnout are concerns for this group as a whole, even in a primarily European sample where MFAs are likely to have more resources to support spouses. Our sample was not large enough to allow comparisons across foreign services; more research is needed to understand this population in more depth.

Furthermore, as with all survey-based research, selection bias may have affected our results. The survey was only available in English, and was conducted during a very stressful time which could have resulted in lower participation by respondents with increased workloads. Alternatively, those under great stress may have felt more urgency to share what they were experiencing. Our cross-sectional study design allowed us to observe some significant correlations, but does not allow us to presume causation.

In assessing organizational support, we asked respondents to indicate which supports they were aware of. There may have been supports offered that respondents were not aware of and, conversely, some respondents may have believed their MFAs offered some supports that, in fact, were not available. We did not attempt to assess the accuracy of these self-reported supports. Future research could assess the quality of communication with foreign service spouses by comparing supports as perceived by spouses with actual support offered by each respondent's MFA. However, as direct supports are unlikely to affect resilience or burnout if partners and spouses are not aware of them, we believe that assessing spouses' perceptions of supports was the right approach in this study. It is also important to recognize that levels and types of support vary greatly from country to country; for example, while some MFAs have a dedicated Family Office to answer questions and support to partners and spouses, the actual staffing, focus, and support provided by these Family Offices varies widely.

We assume that the levels of personal burnout and resilience we measured were affected by the pandemic, and our results should not be considered representative of this population in the absence of a crisis; however, the lack of previous research on resilience and burnout in this population makes it difficult to put our results in context. We expect that foreign service spouses' needs for support will also change once the pandemic is over, as needs during a crisis can reasonably be expected to differ from needs in the absence of one. More research on resilience and burnout in this population is clearly needed, also after the pandemic has subsided, to determine a more normal baseline in this group.

**Author Contributions:** Conceptualization, S.G., M.W.N., J.D.M. and K.L.; methodology, S.G., M.W.N., J.D.M. and K.L.; formal analysis and data curation K.L.; writing—original draft: S.G., M.W.N., J.D.M. and K.L.; writing—review and editing S.G., M.W.N., J.D.M., K.L. and D.M.-W.; project administration S.G. and D.M.-W.; All authors have read and agreed to the published version of the manuscript.

**Funding:** This study was partly funded by the University of Iceland Research Fund.

**Institutional Review Board Statement:** The study did not require ethical approval according to the University of Iceland Science Ethics Committee. However, it follows the scientific ethics rules of the University of Iceland.

**Informed Consent Statement:** Informed consent was obtained from all subjects involved in the study.

**Data Availability Statement:** The data presented in this study are available upon request from the corresponding author.

**Conflicts of Interest:** The authors declare no conflict of interest.

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
