# Peer review of "Burnout and Resilience in Foreign Service Spouses during the Pandemic, and the Role of Organizational Support"

_sustainability, doi:10.3390/su15032435_

Round 1
Reviewer 1 Report
1. Add Research Hypotheses those hypotheses should be consistent with aims mentioned at the end of theoretical background and research aims along with in depth theoretical underpinnings.
2. Add research framework.
3. Add instrument as an appenidx.
4. Authors must explain whether they have chosen standardized regression coefficients or unstandardized regression coefficients in Table 1? Some beta values are so high.
5. Once hypotheses are developed then Justify in discussion section by comparing with past studies.
6. Add managerial Implications
7. Authors have used Copenhagen Burnout Inventory (CBI) a 6-item subscale to measure personal burnout there might be multivariate analysis issues how authors address this issue.
Author Response
Dear Sir/Madam.
Thank you most kindly to take time from your busy schedule to review our paper. We do believe that due to your valuable work the paper has taken a positive change and we do hope that you will agree to our changes and comments.
Kind regards - the authors
Reviewer 2 Report
The subject matter discussed in the article is very important and presented and analysed in a very interesting way.
Despite my high opinion of the presented study, I have a few remarks about the article:
1. In the abstract, the authors should indicate from which countries the foreign service spouses are from.
2. The abstract should indicate the primary method of analysis used.
3. In chapter 2, lines 70-75, the authors define burnout. Due to the galloping development of new technologies, working conditions are changing, and there is a need to keep up with the latest in the profession. The literature provided by the authors defining occupational burnout and its causes is not up to date. Can the authors refer to the latest definitions on this issue?
4. The authors refer to previous studies conducted in different countries, e.g. lines 109-110, line 255. Can the authors relate these cases of occupational burnout to cultural aspects of the perception of work?
5. Information about people completing the questionnaire appears in the Target population section. Are the guidelines on preventing burnout and protecting employees used by the Ministries of Foreign Affairs of the countries mentioned above comparable?
6. There is no information about the sample's representativeness in the Target population section (or anywhere else in the article). In addition, the organization of diplomatic services is not comparable between countries. The working conditions of people employed in them (and the conditions of stay of families) also do not coincide. Therefore, it is difficult to conclude based on the collected surveys about all families of foreign service officers. It would be necessary to carry out the analysis within the diplomatic service of one country. I am asking the authors to complete the information on the sample's representativeness and refer to the heterogeneity of employment conditions in the foreign service officers of the analysed countries.
7. In the table for the fifth model, the degrees of freedom in the F test should be written as (5,415) and not (5.415).
8. In line 209, I propose a notation unified to the one used in table 1, i.e.: p <0.05, ** Significant at p <0 .01, *** Significant at p < 0.001.
9. The authors write (line 156) that they have 26 variables in the study. Up to eleven variables were used in the models. The authors must clarify which variables were used in the models and which in other analyses.
Author Response
Dear Sir/Madam.
We want to thank you most kindly for taking time from your busy schedule to review our paper. We do believe that due to your valuable work the paper has taken a positive change and we do hope that you will agree to our changes and comments.
Kind regards - the authors